# Barefoot Running on Grass as a Potential Treatment for Plantar Fasciitis: A Prospective Case Series

**DOI:** 10.3390/ijerph192315466

**Published:** 2022-11-22

**Authors:** Stephen MacGabhann, Declan Kearney, Nic Perrem, Peter Francis

**Affiliations:** 1EVOLVE Research Group, Department of Health and Sports Sciences, Southeast Technological University (SETU), R93 V960 Carlow, Ireland; 2School of Medicine, College of Medicine, Nursing & Health Sciences, National University of Ireland, H91 TK33 Galway, Ireland; 3NHS Devon, Exeter EX2 5DW, UK

**Keywords:** barefoot, running, plantar fasciitis

## Abstract

Background: Foot characteristics and running biomechanics in shod populations are associated with the aetiology of plantar fasciitis, the most common musculoskeletal disease of the foot. Previous Case reports have demonstrated improvements in the symptoms of plantar fasciitis after a period of barefoot running on grass. Methods: Recreational runners with symptomatic plantar fasciitis were prospectively enrolled into a 6-week grass based barefoot running programme. Duration of symptoms, previous management and current pain scores (NRS, VAS) were recorded at entry. Daily pain scores were recorded during the 6-week period and 12 weeks from entry to the programme. Results: In total, 20 of 28 patients (71.4%) enrolled were included in the analysis. Relative to the entry point, pain at 6-weeks was lower (2.5 ± 1.4 vs. 3.9 ± 1.4, *p* < 0.001) and pain at the 12-week point was lower (1.5 (1.8), *p* = 0.002). 19 out of 20 patients had improved at week-6 (mean ± SD % change in pain score, −38.8 ± 21.5%) and at week-12 (median (IQR) % change in pain score, −58.3 (34.8) %). Conclusion: Barefoot running on grass improved pain associated with plantar fasciitis at the 6-week and 12-week follow up points. This type of barefoot running has the ability to improve symptoms whilst allowing patients to continue running, the intervention may also address some impairments of the foot associated with plantar fasciitis.

## 1. Introduction

Plantar fasciitis is the most common musculoskeletal condition of the foot in the general population and in the running community [1,2,3]. Mounting evidence suggests that it is caused by repetitive tensile loads applied to the plantar aponeurosis due to excessive deformation of the foots longitudinal arch [2,3,4,5]. Excessive deformation of the arch is facilitated by weakness of the intrinsic foot muscles [2,3,4,6,7]. Modern footwear use is associated with weaker intrinsic foot muscles [7,8] a greater prevalence of flat feet [9] and changes to the shape of the foot that are not seen in habitually barefoot or minimally shod populations [7,10,11]. 

The field of evolutionary medicine has begun to include plantar fasciitis in the category of ‘mismatch diseases’ or ‘diseases of modernity’ [12]. The basis for this categorization comes from the fact that humans and more specifically, the human foot evolved to walk and run barefoot on a variety of natural surfaces over the course of millions of years [13]. The relatively recent invention of footwear (especially modern footwear), combined with the relatively recent invention of artificial surfaces represents a rapid change in the physical environment experienced by the foot. This is much in the same way that high energy availability and an increase in sedentary lifestyles represents a rapid change in our metabolic environment which contributes to diseases associated with metabolic syndrome. Therefore, it may be reasoned that the consistent failure of modern musculoskeletal medicine to prevent or improve the symptoms associated with plantar fasciitis may in part be attributed to approaches that lack of consideration for the human body within an evolutionary context [14,15]. The failure to achieve optimal outcomes in those suffering with plantar fasciitis is especially true in runners who experience a median recovery time of 5 months [16] Conventional treatment of plantar fasciitis includes, but is not limited to, manual therapy, stretching [17], taping [18], dry needling of myofascial trigger points in the triceps surae region [19], proximal and distal lower limb muscle strengthening [20], orthotics and nights splints [21], shockwave therapy and injection therapies [22]. While all have shown promise in terms of short-term symptom modulation, none have demonstrated long term effectiveness that could inform clinical practice guidelines [14]. In fact, 54% of all patients followed for ~10 years reported symptoms of ~2 years in duration and the remainder were still symptomatic at follow up [22].

Applying an evolutionary lens to our clinical practice led to us guiding an athlete to run her way out of plantar fasciitis (on-going for 1-year) via removing her footwear and running on a natural (grass) surface for 15 min every other day. This led to an immediate resolution in symptoms that were maintained at the 7-week follow-up point [23]. Following similar progress in a small number of athletes [24], the purpose of this study was to investigate the effect of barefoot running on grass to reduce the symptoms of plantar fasciitis in a patient case series. This paper presents data on 20-runners with diagnosed and symptomatic plantar fasciitis who undertook a barefoot running intervention on grass whilst having daily pain scores monitored for 6-weeks and at a singular follow-up conducted at 12-weeks.

We hypothesised, that despite regular running, pain scores would fall when running was conducted barefoot on grass by the end of the intervention (week 6) and remain reduced at the 12 week follow up point.

## 2. Materials and Methods

A convenience sampling method utilising a combination of social media and email correspondence with running clubs in the Republic of Ireland was used to issue an open call for runners (≥18 years) with a confirmed medical diagnosis of plantar fasciitis to volunteer to take part. Volunteers who expressed an interest received a telephone call from a medical practitioner (lead author) to re-confirm a diagnosis of plantar fasciitis using published guidelines in relation to signs and symptoms [21]. This phone call was also used to verify if the volunteer was a recreational runner (running a minimum of 5 km every week prior to or during injury). Volunteers who met the criteria of being a recreational runner with existing symptomatic plantar fasciitis were provided with a participant information sheet and invited to provide written informed consent. The flow chart of volunteers assessed for eligibility, participants recruited, participants who dropped out and those included in analysis is displayed in Figure 1.

Prior to undertaking the barefoot running intervention, participants were asked to provide information on the duration of their plantar fasciitis and on any previous treatments obtained along with the associated cost of treatment. Participants were asked how plantar fasciitis had affected them and their responses were categorised into: requiring cessation of activity, significant pain or disability and no interruption to daily life/unknown impact. Participants were also asked to indicate how they felt about the prospect of running barefoot on grass during the first week of the intervention. These responses were coded as neutral, apprehensive or positive.

Each participant received an independent follow up phone call explaining the ‘barefoot running’ process to ensure there was no misinterpretation. A briefing regarding the possible dangers associated with running barefoot on grass was also discussed. This included objects on the surface of grass playing fields along with animal faeces. Participants were asked to run barefoot on grass for 15 min at an intensity that represented a rate of perceived exertion (RPE) of 11 every 2nd day for 6-weeks (21-sessions). A minimum of 2 sessions per week for 6-weeks was required to be considered compliant. Morning pain scores were recorded everyday using a Numeric Rating Scale (NRS) accompanied by a Visual Analogue Scale (VAS) [25,26] All participants were informed of the interventional protocol and were requested not to seek any other form of treatment for the duration of the intervention but were encouraged to carry out their normal activities of daily living including their current exercise regime.

Each participant was requested to complete a daily online survey which consisted of two questions and an option to leave any feedback they felt was relevant. Question One: What activity did you perform yesterday (if other please specify activity and duration in time and distance)? Option A—Rest Day; Option B—Ran 15 min on grass barefoot; Option C—Other. Question Two: How would you describe your pain first thing this morning using the Numeric Rating Scale? Zero is no pain at all. Ten is the worst pain imaginable. Each participant was sent a link to access the survey daily. A reminder was sent to individuals that failed to fill in the daily survey by 21:00 GMT by SMS text message. If a participant failed to fill out the survey on two consecutive days, they were excluded from the study.

At week 6, participants were given the option of continuing the intervention and recording the online survey. A follow up on week 12 was conducted to assess the following: (a) morning pain score, (b) if participants continued with the barefoot intervention after the monitoring period ceased, (c) whether they would recommend this intervention to someone else with their condition and (d) whether they had returned to running in shoes at least twice per week for a minimum of 5 km. These data were collected on a randomly assigned day during the 12th week after the study commenced.

This study was approved by the College of Medicine and Health Sciences Research Ethics Committee at the National University of Ireland, Galway.

Daily survey results were collected on a cloud-based storage drive, data were transferred into Microsoft excel and prepared for analysis in IBM SPSS (version 25). Data were checked for normality of distribution using the Shapiro–Wilk test. Normally distributed data were expressed as mean ± standard deviation (SD) with 95% confidence intervals (CI), median, interquartile range (IQRs) and bootstrap 95% CI was calculated for non-normally distributed variables. A repeated measure (within subject) analysis of variance was used to determine the effect of time on mean pain scores between weeks 1–6 and 12. Pairwise comparisons were used to determine at what time points differences occurred. Individual percentage change was calculated between each time point and base line and the group mean displayed to represent overall relative percent change in pain score. The a value for the acceptance of statistical significance was set at set at *p* < 0.05, with a Bonferroni correction applied for multiple pairwise comparisons.

## 3. Results

20 participants (male, n = 10 and female, n = 10) aged 34–65 years (mean ± SD, 48 ± 8) completed the full intervention and were included in the analysis. Participants had been experiencing symptoms associated with plantar fasciitis for between 3 and 48 months (median ± IQR, 7.5(12)). Cessation of activity (n = 6, 30%) or significant pain and or disability (n = 12, 60%) was reported by 18 participants, one participant reported no interruption to activity and one did not answer the question. The most common prior treatments obtained by participants were physiotherapy (n = 9, 45%) and shockwave therapy (n = 2, 10%). 40% (n = 8) of the cohort had not received prior treatment. For those who had received treatment, the costs incurred were as follows, ≤EUR 200 (n = 8), EUR 300–EUR 500 (n = 2) and EUR 500–EUR 1000 (n = 2). The majority of the sample were neutral (n = 11) about the prospect about taking part in the intervention and the remaining participants were apprehensive (n = 4) or positive (n = 5). Compliance to the intervention was 76.2% (16.1 ± 3.0 running sessions, maximum = 21). Compliance to 3 days per week was 88.9%. Table 1 displays NRS pain scores for weeks 1–6 and the follow up at week-12.

The mean or median relative percentage change in pain score from baseline is displayed in Figure 2. The difference from base line at week 2 (−5.1 ± 26.6%), week 3 (−4.8 ± 28.2%), week 4 (−11.2 ± 34.5%), week 5 (−21.4 ± 27.4%), week 6 (−38.8 ± 21.5%) and 12 (−58.3 ± 34.8%) are displayed, respectively.

Figure 3 displays mean or median and individual relative percent change in pain scores at the end point of the study (week-6) and at the 12-week follow up point. Mean and median pain scores were statistically lower at week-6 (−38.8%, 95% CI: −28.7–−48.8%, range: −85.4–3.7%) and week-12 (58.3%, 95% CI: −40.0–−96%, range −100–96%), respectively.

Eighteen out of 20 participants would recommend the intervention to someone else with their condition. Twelve of the participants were running at least 5 km twice weekly, in shoes and on the road at the 12-week point. One participant continued to run barefoot on the grass at the 12-week point.

## 4. Discussion

Plantar fasciitis is associated with increased time spent standing, walking and running, particularly on unyielding surfaces [27,28]. Individuals with a greater mass or those with ‘heavier’ running mechanics may be particularly at risk [27,29]. The symptoms of plantar fasciitis are most noticeable after prolonged activity or during the first steps after a prolonged period of rest, i.e., when seated or sleeping [21]. Therefore, it is highly counterintuitive to both patients and clinicians to prescribe or adopt vigorous loading activities (running) on a regular basis and without shoes when symptomatic. However, our case series demonstrates that this approach in combination with a natural pliable (grass) surface is not only well tolerated but appears to reduce pain whilst maintaining or increasing running load. We acknowledge that evidence from case series does not allow for the appraisal of efficacy for an intervention. Understanding efficacy would require a well-controlled experimental study design to be utilized. The comparatively novel nature of this intervention for this condition does not make it well suited to a randomised clinical trial until a case series and further observational research has been undertaken. This case series is prone to the same limitations as all case series, the absence of a control group and the random exposure of individuals to the treatment of interest. Furthermore, this case series has an absence of both investigator and participant blinding in relation to the intervention and the changes in the variable of interest. We are also unsure of what impact the passage of time without intervention would have had on the changes observed in pain scores or had the participants continued to run in their shoes on hard surfaces. Additionally, whilst all participants included in the case series were habitually shod, the time spent barefoot whilst completing other activities was not record. The time spent barefoot in other activities may have a confounding effect on the observation made.

These limitations acknowledged; this data should spark further investigation into similar treatment methods for plantar fasciitis. We make this suggestion based on the following observations:(a)the existing outcomes associated with the conventional management of plantar fasciitis are poor(b)the median recovery time in runners is 5 months [25] and can be 2–10 years in the general population [12](c)the outcomes achieved in this case series occur in as little as 1.5 months(d)the uniform (19 out of 20 runners) improvement in the sample studied at 6-weeks and 12-weeks (18 out of 20 runners)(e)the improvement in the condition despite using the activity (running) that often brought on the condition(f)the known changes in intrinsic foot muscle morphology that occurs due to altered kinematics and kinetics associated with this type of training(g)the patient satisfaction that is experienced through immediately returning to their preferred activity as part of the rehabilitation process. This may also help to increase compliance to other aspects of advice and rehabilitation offered by the clinician.

Although these results are encouraging and worthy of further investigation, explaining why they might occur is not an easy task. Based on the introduction to this manuscript it is tempting to assume that restoration of intrinsic foot muscle size and strength which better serves the integrity of the arch and therefore, reduces the repetitive tensile loading to the plantar fascia is the reason for the improvements shown. We suggest that this plays a role in the results observed at week-6 and 12, but our experience with some individuals show an almost instantaneous resolution of symptoms that cannot be explained by changes in muscle size and strength [24]. To address the short and longer term effects of this intervention on pain in runners, we have split the discussion into short-term and long-term.

### 4.1. Short-Term Factors That May Alter Pain Sensitivity

The loading impulse experienced during walking is 2–3 greater in footwear compared to walking barefoot [30]. Runners who rearfoot or midfoot strike (most shod runners) experience greater tibial shock with increasing running speed relative to forefoot strikers during a marathon [31]. Higher impact kinetics, such as these, are associated with the most common running injuries [1,32]. Most runners alter running kinematics and kinetics when first exposed to barefoot running [33]. Therefore, it is possible that the mere introduction of change in relation to running biomechanics could introduce sufficient variability into overused patterns and or lower impact kinetics enough to create immediate pain change. Linked to this explanation is the runner’s acute exposure to a new foot-surface environment via the pliable (grass) surface used. Plantar pressure and external ankle joint moments are lower running on a pliable surface relative to firm surfaces [34,35]. This is coupled with the knowledge that runners, whether habitually shod or barefoot, use a more varied foot-strike pattern on soft surfaces [36]. This type of running may allow runners to capitalise on the inherent variability of the foot (33-joints) and the natural surface to produce a consistent action (running) via different patterns of tensioning, stiffening and joint relations. Indeed, the energy return from connective tissues associated with the medial longitudinal arch [2], transverse arch [37] and heel [38] make a significant contribution to energy turnover during running [4]. It may be that liberating the foot from the shoe and the almost permanent state of ‘windlass’ associated with the upward curvature of toe springs [2] allows runners make greater immediate use of their foot arches, thereby reducing overload to the plantar fasciitis and the symptoms associated.

It is important to consider sources of short-term analgesia that may be less foot specific. Maximal isometric contractions of the triceps surae and quadricep femoris muscles can lead to immediate pain relief in patients suffering from achilles and patella tendinopathy, respectively [39]. Given that intrinsic foot muscles such as the abductor hallucis and flexor digitorum brevis act as arch stabilisers and are most active during dynamic movements [2,7] the increased isometric muscle work may provide an analgesic benefit, although this suggestion remains speculative. Novel stimulation of the skin and muscles in this way may also provide an analgesic effect much in the same way most sensory based therapies are suggested to via a pain gate mechanism [40]. Finally, although our participants were not informed about any potential responses to the intervention, if participants suspected they were engaging in a novel activity that might be of benefit, psychological mechanisms could have contributed to some of the acute pain changes seen [41].

### 4.2. Longer-Term Factors That May Alter Pain Sensitivity

Healthy feet (i.e., not flat footed) are characterised by stiffer arches [7] that undergo less deformation during dynamic activities as a result of strong intrinsic foot muscles functioning in concert with passive tissue restraints [38]. This is evident from habitually barefoot populations having a lower prevalence of flat foot and larger abductor hallucis and abductor digit minimi muscles [7,9,42,43]. Furthermore, shod populations who demonstrate the reverse in terms of arch integrity and foot muscle morphology, can develop functional foot characteristics more in likeness to habitually barefoot populations after a period of barefoot or minimally shod walking and running [8,44]. It may be these adaptations that have contributed to the sustained reduction in pain in the runners in this study at 6 and 12-weeks. This explanation is supported by the findings of Cheung et al. [6] who report runners with chronic plantar fasciitis to have smaller intrinsic foot muscles than their healthy counterparts.

The coordinated action of the neural, passive and muscular subsystems is thought to underpin healthy movement that does not overload any one tissue [4,5]. Failure in any one of these subsystems is likely to disrupt coordinated movement. Restoration of foot muscle function and its integration with the passive subsystem, combined with increased sensory input to the skin may have played a role in the positive pain changes observed at week-6 and 12. Evidence in support of this suggestion has been provided by Venkadesan and colleagues [38] who have highlighted the important role of the transverse arch in human foot stiffness. The authors suggest that muscles such as the tibialis posterior which are highly active in mid stance before relaxing in late stance provide an example of muscular and passive subsystems working in tandem to allow recovery of elastic energy. Intrinsic foot muscle dysfunction (resulting in excessive arch deformation) may be one of the reasons that muscles such as tibialis posterior and passive tissues such as the plantar fascia become overworked in shod runners [28,45].

Differences in ankle and foot dynamics will influence the entire kinetic chain. Runners who respond positively to a barefoot running training programme demonstrate increased pre-activation of the gluteus medius and biceps femoris muscles and a reduction in rectus femoris activation [46]. It may be that runners with the biomechanical and physiological characteristics (kinematics, kinetics and foot muscle dysfunction) associated with plantar fasciitis respond especially well to this type of barefoot running. This may also contribute to the uniform responses observed in this study.

### 4.3. Other Considerations

This intervention commenced in the spring of 2020, shortly after the onset of the global pandemic. This time period was associated with unseasonably warm weather in Ireland which led to firmer than normal ground. Based on our previous clinical experience, changes in symptoms may have occurred sooner had the surface been more pliable, as it was midway and toward the end of the intervention. Two participants did not report improvements at the 12-week point. Both were followed up. One participant had exercised (personal regime) to the point of plantar fasciitis tear and had ceased running. The second participant had increased their training load but also noted that they felt the firm ground had not helped them. With these considerations in mind, it is important to recognise that this intervention may not work for all patients and that it should be carefully progressed on an appropriate surface, much like any exercise intervention.

## 5. Conclusions

Efficacious outcomes are not always achieved when conventional approaches to injury management are used for patients with plantar fasciitis. Mounting evidence supports the thesis that plantar fasciitis maybe a mismatch disease associated with modernity (modern footwear-surface use). Reconceptualising this injury using an evolutionary medicine perspective allows for greater consideration to be given to the impact of modern footwear on factors associated with the condition such as foot morphology. Changes in running biomechanics and foot morphology have been associated with barefoot running. Barefoot running on grass appears to be well tolerated by runners with symptomatic plantar fasciitis. Within this case series pain associated with plantar fasciitis appeared to become lower while running load increased in a group of recreational runners completing a barefoot running intervention. This case series provides results which may be used to inform further clinical trials or to underpin the clinical reasoning used by applied sport and exercise medicine practitioners.

## Figures and Tables

**Figure 1 ijerph-19-15466-f001:**
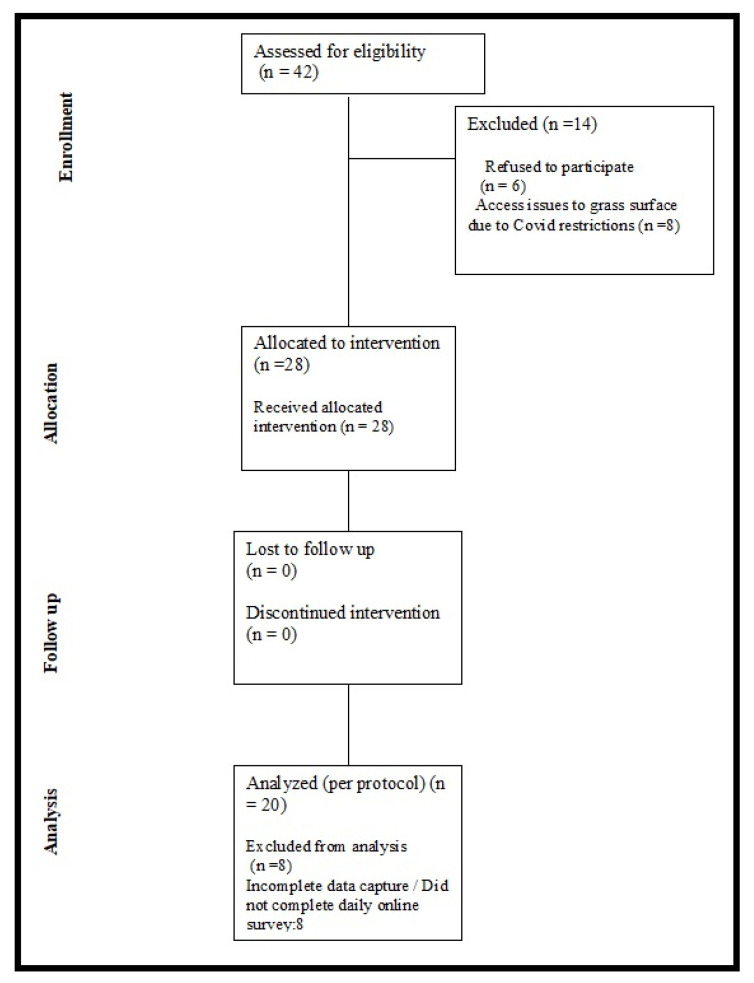
Participant recruitment and participation flow chart.

**Figure 2 ijerph-19-15466-f002:**
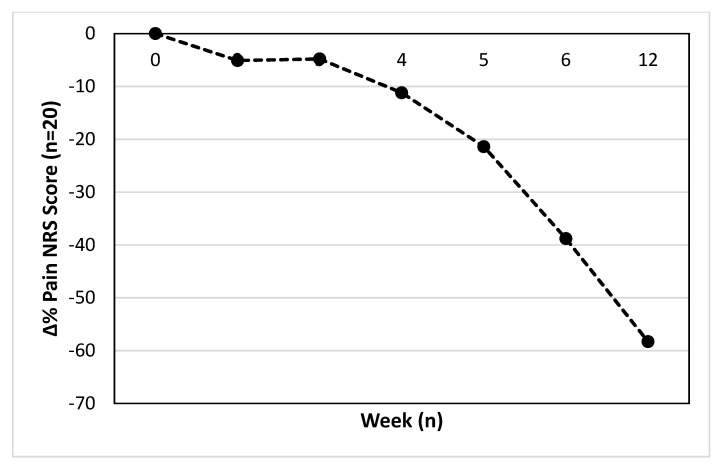
Relative percentage change in mean (week 2–6) or median (week 12) pain score from week 1.

**Figure 3 ijerph-19-15466-f003:**
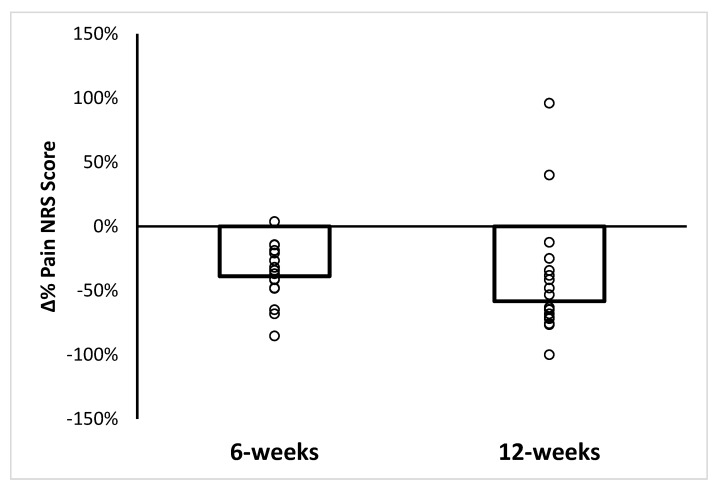
Individual and mean (week 6) or median (week 12) relative percentage in pain score.

**Table 1 ijerph-19-15466-t001:** Weekly pain scores during the intervention and at the 12-week follow up point.

	Week-1	Week-2	Week-3	Week-4	Week-5 *	Week-6 *	Week-12 *
Mean ± SD	3.9 ± 1.4	3.7 ± 1.4	3.7 ± 1.4	3.5 ± 1.6	3.2 ± 1.8	2.5 ± 1.4	1.5 (1.8)
95% CI	3.3–4.6	3.0–4.3	3.0–4.3	2.8–4.3	2.4–4.0	1.9–3.2	1.0–2.0
Range	1.7–7.1	1.3–7.3	1.0–6.9	0.4–6.4	0.7–7.6	0.3–5.6	0.0–7.0
*p*-value		0.209	0.275	0.125	0.003	<0.001	0.002

SD = standard deviation; CI = confidence interval; ***** = statistical difference in pain score from week 1 using corrected *p*-value of <0.008 from Bonferroni correction.

## Data Availability

The data presented in this study are available on request from the corresponding author. The data are not publicly available due to ethical considerations.

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
