# Peer review of "Barefoot Running on Grass as a Potential Treatment for Plantar Fasciitis: A Prospective Case Series"

_ijerph, 2022, doi:10.3390/ijerph192315466_

Round 1

Reviewer 1 Report

The introduction does not adequately explain the main biomechanical cause of plantar fasciitis: shortening of the gastrocnemius and/or hamstrings. The literature cited is biased because only the issue of wearing footwear was explored, generating weakness of the intrinsic musculature of the feet. There is no plausible justification presented for running barefoot to have any benefit, since a person does not walk barefoot all day.

The methodology is not adequate as it was based only on a telephone interview and an online questionnaire. Physical examination is essential for gait assessment: foot type (flat, cavus or regular), quantification of lower limb muscle strength (with a dynamometer) including intrinsic muscles, foot and ankle/knee/hip range of motion and lower limb alignment.

The biomechanical assessment of running with and without shoes needs to be performed, and in order to avoid bias, the same type of shoes must be used in all patients undergoing the study.

The paper has a serious methodological flaw from the beginning, so results and discussion cannot be considered since they start from a false premise.

Reviewer 2 Report

The authors cannot really answer their main question with their current design.

Thank you for allowing me to review “Barefoot running on grass is a potential treatment for plantar fasciitis: A case series.” The authors aim to test if running barefoot is associated with a decrease in plantar fasciitis symptoms. Given the prevalence of the plantar fasciitis and the relative ease of implementing this intervention, the results are potentially of high impact. Overall,

Abstract: The end of the abstract is cut off. The authors should double check the word limits of the journal.

Introduction: Very clearly written and theoretically-informed. The summary of the literature is appropriate.

Methods: The methods are very problematic, most notably because there was no random assignment nor a control group. Ideally, participants would have been randomly assigned to run with shoes or to run barefoot. A case series design is not appropriate to answer the intended question. The authors acknowledge this as a limitation, but they do not explain why they did not do this.

A second major limitation of the methods section is that the authors did not control how much people regularly are in bare feet in their everyday activities. If people are barefoot for much of the day (ie. At home, in their free time, etc) then that may equally as beneficial as running barefoot. Without controlling or assessing what people’s regular barefoot patterns are, it is very hard to say if the effect was due to running barefoot.

Overall, the manuscript is clearly written and the results are somewhat encouraging, although difficult to make sense of. If the authors could add a control group it would make the paper much more compelling (even if it is too late to randomly assign people to condition).

Reviewer 3 Report

A very good study, but few corrections are needed. You may find my comments in the attached manuscript.
